# S and Sr Isotope Compositions and Trace Element Compositions of the Middle Jurassic Evaporites in Eastern Tibet: Provenance and Palaeogeographic Implications

**Jinna Fei [1], Lijian Shen [2],\*[ID], Xin Guan [3] and Zhicheng Sun [4]**

[1]  School of Earth Sciences and Resources, The China University of Geosciences, Beijing 100083, China
[2]  MNR Key Laboratory of Metallogeny and Mineral Assessment, Institute of Mineral Resources,
    Chinese Academy of Geological Sciences, Beijing 100037, China
[3]  China Energy Investment Corporation, Beijing 100011, China
[4]  School of Water Resources and Environment, The China University of Geosciences, Beijing 100083, China
**\***  Correspondence: shenlijian@cags.ac.cn

**Abstract:** The origin of Middle Jurassic evaporites in the Qamdo Basin is still controversial because palaeontological studies have reported that they have both marine and continental characteristics. The $^{87}Sr/^{86}Sr$ ratios of the gypsum in the Middle Jurassic Dongdaqiao Formation in the Qamdo Basin range from 0.707602 to 0.708163, which are higher than that of contemporaneous seawater. Model calculations suggest that continental water prevailed over seawater during the precipitation of these evaporites. However, the majority of the gypsum samples have $\delta^{34}S$ values of 15.3‰ to 16.3‰, which are consistent with that of contemporaneous seawater. This range of values (15.3‰ vs. 16.3‰) was likely caused by S isotope fractionation during evaporation because the $\delta^{34}S$ values and Sr contents are negatively correlated. The $\delta^{34}S$ values of the other three gypsum samples are 20.0‰, 20.5‰, and 20.8‰, which are significantly higher than that of Middle Jurassic seawater. The trace element compositions and scanning electron microscopy (SEM) observations indicate that these elevated $\delta^{34}S$ values were caused by bacterial sulphate reduction (BSR). The Sr and S isotope systematics of the gypsums from the Dongdaqiao Formation demonstrate that the parent brines from which the evaporites precipitated were marine based with a large quantity of continental input. A comparison of the lithologies and Sr isotope compositions of the Middle Jurassic sequences in the Qamdo and Qiangtang Basins revealed that the Qiangtang Basin was mainly recharged by Jurassic seawater, while the Qamdo Basin was primarily recharged by continental water with some seawater-derived overflow from the Qiangtang Basin.

**Keywords:** the Qamdo Basin; gypsums; S and Sr isotopes; trace elements; provenance and palaeogeography

## 1. Introduction

The Qiangtang Basin is a large Mesozoic marine sedimentary basin located on the northern Qinghai-Tibet Plateau [1,2]. Evaporites, mostly anhydrites with subordinate halites, are widely distributed in the Middle Jurassic Formations within the middle part of the Qiangtang Basin [3]. The eastern part of the Qiangtang Basin, i.e., the Qamdo Basin, contains Middle Jurassic gypsums, which could be correlated with the middle part of the Qiangtang Basin. The Qiangtang and Qamdo Basins have been regarded as an integrated basin in the Qiangtang-Qamdo block since the Middle Jurassic period [4]. However, the diversity between the Qiangtang and Qamdo Basins with respect to the palaeontology of Middle Jurassic sedimentary rocks shows distinct features. Marine fossils, including bivalves, brachiopods, gastropods, corals, echinoderms, etc., from Middle Jurassic sedimentary rocks of the Qiangtang Basin, are characterised by a narrow salinity range, denoting epicontinental sedimentation environment [5]. Conversely, bivalve assemblage within the Middle Jurassic Dongdaqiao Formation in the Qamdo Basin indicates both

marine and nonmarine characteristics, and the Dongdaqiao Formation contains continental vertebrates and plant fossils [6]. The discrepancy of different sedimentary environments reflected by bivalve assemblage and continental vertebrates and plants of the Dongdaqiao Formation was likely caused by a complex sedimentary process influenced by marine and non-marine inputs. Studies on drilling cores in the eastern part of the Qiangtang Basin showed a marine to nonmarine transformation from Triassic to Early-Middle Jurassic [7]. Thus, there is an active debate regarding the origin of Middle Jurassic evaporites in the Qamdo Basin.

Sr and S isotopes of sulphate minerals could be used to determine the origin of dissolved sulphate in the brines that precipitate those minerals [8]. The Sr and S isotope compositions of marine evaporites are well constrained through the Phanerozoic [9–12]. In comparison to marine evaporites, these isotopic compositions of continental evaporites are more complex, depending on local geology and hydrology within the drainage basin [13]. In this paper, we present Sr, S isotopes, and trace elements of gypsums from the Dongdaqiao Formation, aiming to determine the origin of parent brines in which evaporites precipitated. In addition, the brine origin can provide information for the evolution of depositional history [14]. Based on the geochemical framework for the Qamdo and Qiangtang Basins, we also discuss the implications for the palaeogeographic features of the Middle Jurassic sedimentation in the Qamdo and Qiangtang Basins.

## 2. Geologic Setting

The Qamdo Basin is located on the eastern part of the Qinghai-Tibet Plateau (Figure 1), covering an area of about $4.8 \times 10^4$ km$^2$. The basin mainly developed within the North Qiangtang-Qamdo-Simao Block, which is bounded by the Jinshajiang suture to the northeast and the Longmu Co-Shuanghu suture to the southwest (Figure 2A) [15]. The Jinshajiang and Longmu Co-Shuanghu sutures record different branches of the Palaeo-Tethys, which opened in the Early Devonian and closed in the Triassic [16].

The exposed gneisses and granulites within the basin indicate the presence of a Pre-Cambrian (Middle Neoproterozoic) crystalline basement. The Lower and Middle Ordovician strata consist of mixed carbonate-siliciclastic shallow marine sediments with upward shallowing cycles. The Silurian sediments developed in various palaeogeographic settings and depositional environments, including coastal sandbars, restricted embayment, lagoon, and open platform environments. The major fluvial and coastal sedimentary facies deposited in the Early Carboniferous evolved into carbonate platform facies during the Middle-to-Late Carboniferous. Early Permian open-platform sediments are composed of mixed carbonate-siliciclastic rocks. The overlying Late Permian formation is a shallowing-upward sequence composed of fine clastic rocks, coal-bearing clastic rocks, and carbonates. During the Early Triassic, sedimentation only occurred in local depressions. A suite of clastic-carbonate-intermediate volcanic rocks was deposited in the sags flanking the block during the Middle Triassic. The occurrence of carbonate platform sedimentation indicates that a major marine transgression occurred during the Late Triassic. Jurassic-Cretaceous red beds are widely distributed in the basin [17]. The Middle Jurassic sedimentary rocks only consist of the Dongdaqiao Formation (Figure 2B), which is composed of thickly bedded purple fine-grained sandstones intercalated with muddy purple siltstones and fine quartzose sandstones. Local gypsum layers or lenses occur in the Dongdaqiao Formation (Figure 3, [18]).

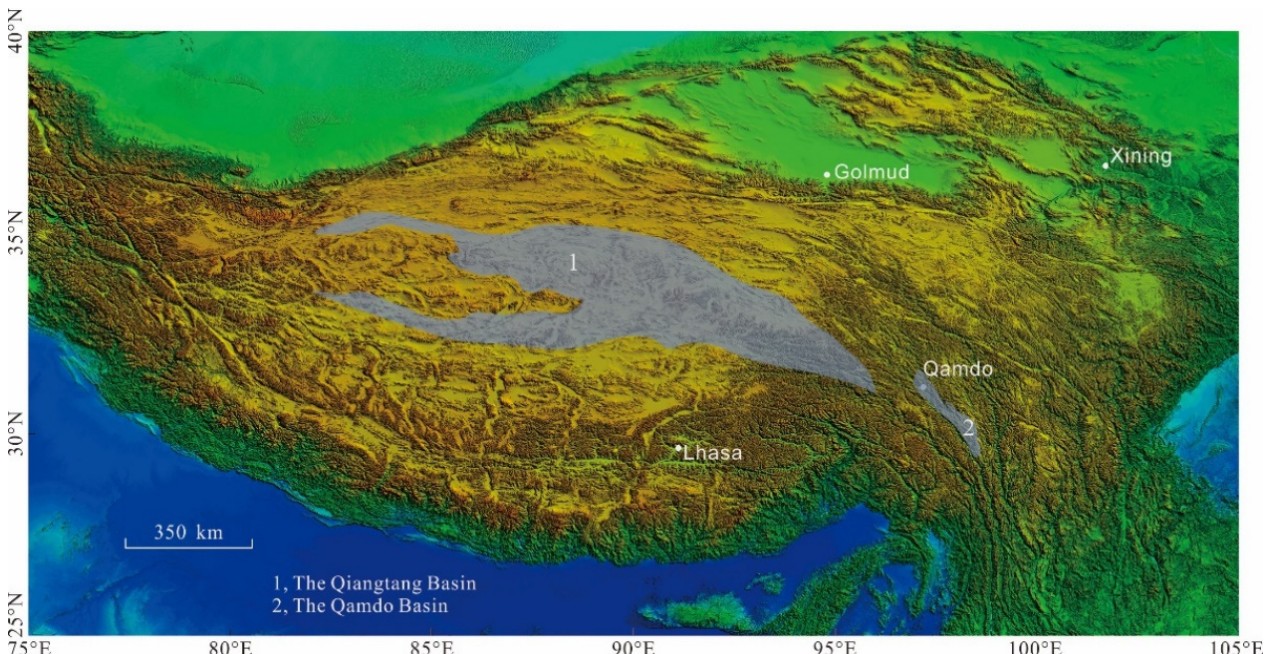

**Figure 1.** The distribution of Qiangtang and Qamdo Basins (modified from [19]).

As the counterpart to the Middle Jurassic Dongdaqiao Formation in the Qamdo Basin, the Middle Jurassic sedimentary sequences in the Qiangtang Basin consist of the Quemocuo, Buqu, and Xiali Formations. The Quemocuo Formation consists of sandstone and mudstone with horizontal bedding, parallel bedding, and cross-bedding. The Buqu Formation was mainly composed of carbonates with horizontal bedding. The Xiali Formation is mainly composed of sandstone, siltstone, and mudstone intercalated with carbonates, muddy carbonates, and anhydrite layers (Figure 3) [20,21]. The anhydrite layers mainly occur in the Xili Formation and subordinately in the Quemocuo and Buqu formations.

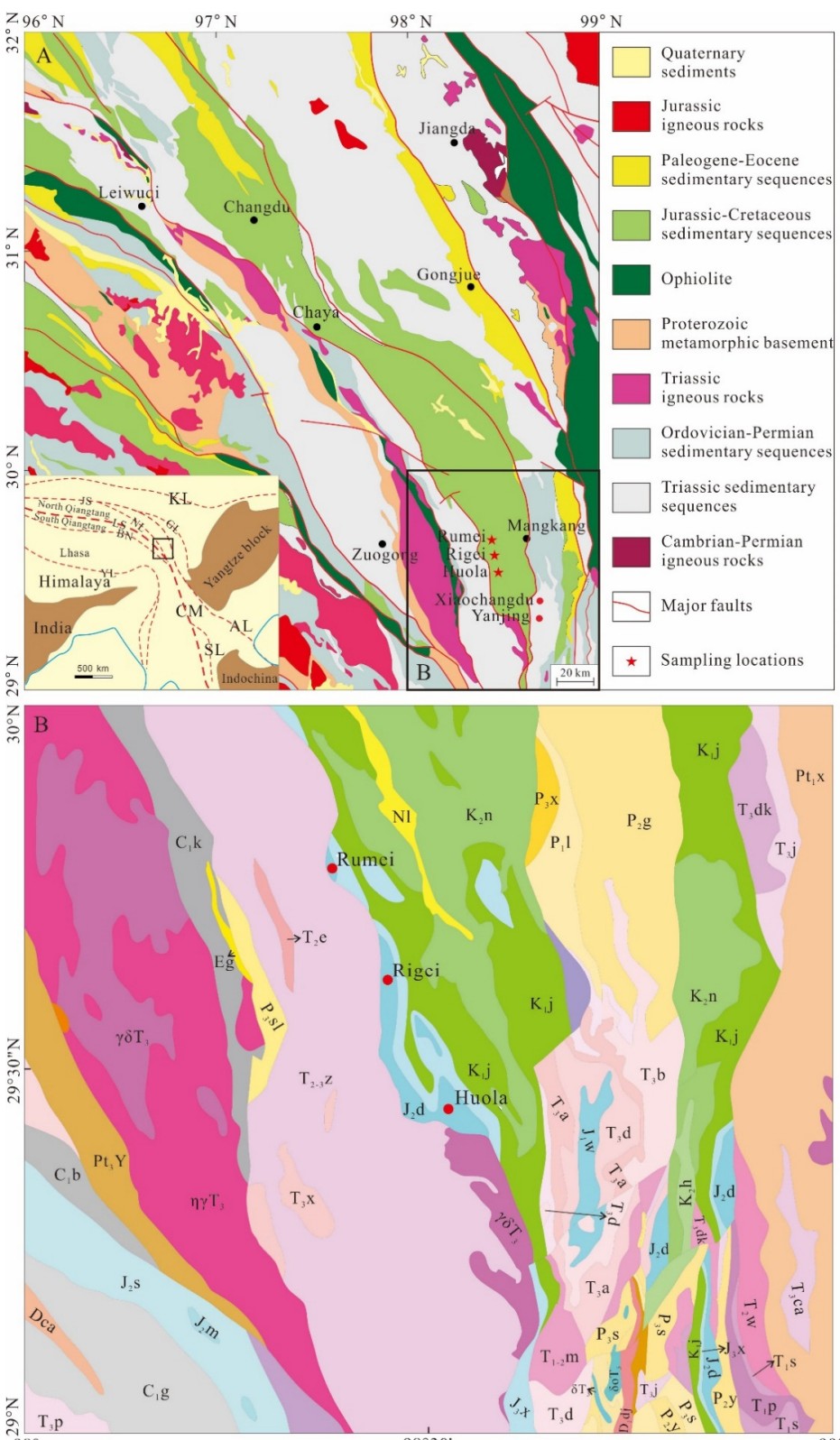

**Figure 2.** The schematic geologic map of the Qamdo Basin (**A**, Modified from [15]), and the distribution of the Dongdaqiao Formation (J₂d) (**B** modified from [6]). KL: East Kunlun-A'nyemaqen suture, GL: Ganzi-Litang suture, JS: Jinshajiang suture, AL: Ailaoshan suture, NL: North Lancangjiang suture, SL: South Lancangjiang suture, LS: Longmu co-Shuanghu suture, CM: Changning-Menglian suture, BN: Bangong-Nujiang suture, YL: Indus-Yarlung Tsangpo suture.

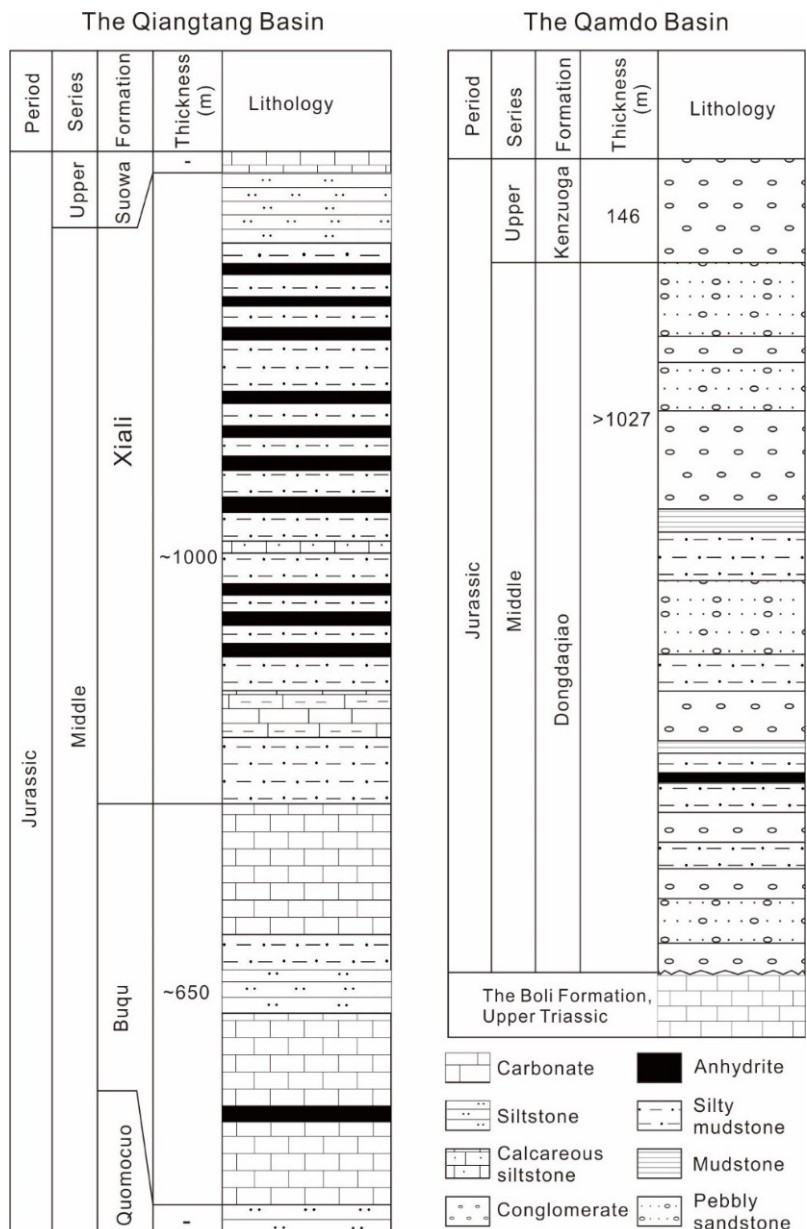

**Figure 3.** The lithostratigraphies of Middle Jurassic sedimentary sequences from the Qiangtang and Qamdo Basins (Modified from [19,21]).

## 3. Materials and Methods

Sixteen gypsum samples were collected from Dongdaqiao Formation outcrops in Rumei (4), Rigei (4), and Huola (8) (Table 1, Figure 2). The samples from Rumei were collected from lenticular gypsum intercalated with purplish sandstone (Figure 4a). Unconsolidated gypsum was mixed with carbonate breccias (Figure 4b). The gypsum crystals from Rumei were subhedral and amoeboid in shape (Figure 4c). The samples from Rigei were collected from well-bedded gypsum intercalated with brown siltstone (Figure 4d). The gypsum layers were characterised by the number of detrital components (Figure 4e). The gypsum crystals collected from Rigei were larger than those collected from Rumei (Figure 4f). Millimetre-scale gypsum layers (gypsum laminae) occurred in Huola (Figure 4g,h). The very fine gypsum grains were linearly aligned (Figure 4i).

**Table 1.** Description of Middle Jurassic sulphates from the Qamdo Basin.

| Location | Sample ID | Age | Formation | Lithology |
|---|---|---|---|---|
| Huola | MKHL-20 | Middle Jurassic | Dongdaqiao | Outcropped gypsum laminae occurred within clastic rocks of Dongdaqiao Formation with corroded surface. Gypsum crystals are subhedral to euhedral and aligned linearly. |
|  | MKHL-21 | Middle Jurassic | Dongdaqiao |  |
|  | MKHL-22 | Middle Jurassic | Dongdaqiao |  |
|  | MKHL-23 | Middle Jurassic | Dongdaqiao |  |
|  | MKHL-24 | Middle Jurassic | Dongdaqiao |  |
|  | MKHL-25 | Middle Jurassic | Dongdaqiao |  |
|  | MKHL-26 | Middle Jurassic | Dongdaqiao |  |
|  | MKHL-27 | Middle Jurassic | Dongdaqiao |  |
| Rigei | MKRG-28 | Middle Jurassic | Dongdaqiao | Alternating white and brown gypsum layers, which are mainly composed of subhedral-to-euhedral relatively coarse grains. |
|  | MKRG-29 | Middle Jurassic | Dongdaqiao |  |
|  | MKRG-30 | Middle Jurassic | Dongdaqiao |  |
|  | MKRG-31 | Middle Jurassic | Dongdaqiao |  |
| Rumei | MKRM-33 | Middle Jurassic | Dongdaqiao | Lenticular gypsum interbedded with purplish sandstone, and massive unconsolidated gypsum mixed with carbonates breccias. Amoeboid gypsum crystals show subhedral-to-euhedral characteristics. |
|  | MKRM-34 | Middle Jurassic | Dongdaqiao |  |
|  | MKRM-35 | Middle Jurassic | Dongdaqiao |  |
|  | MKRM-36 | Middle Jurassic | Dongdaqiao |  |

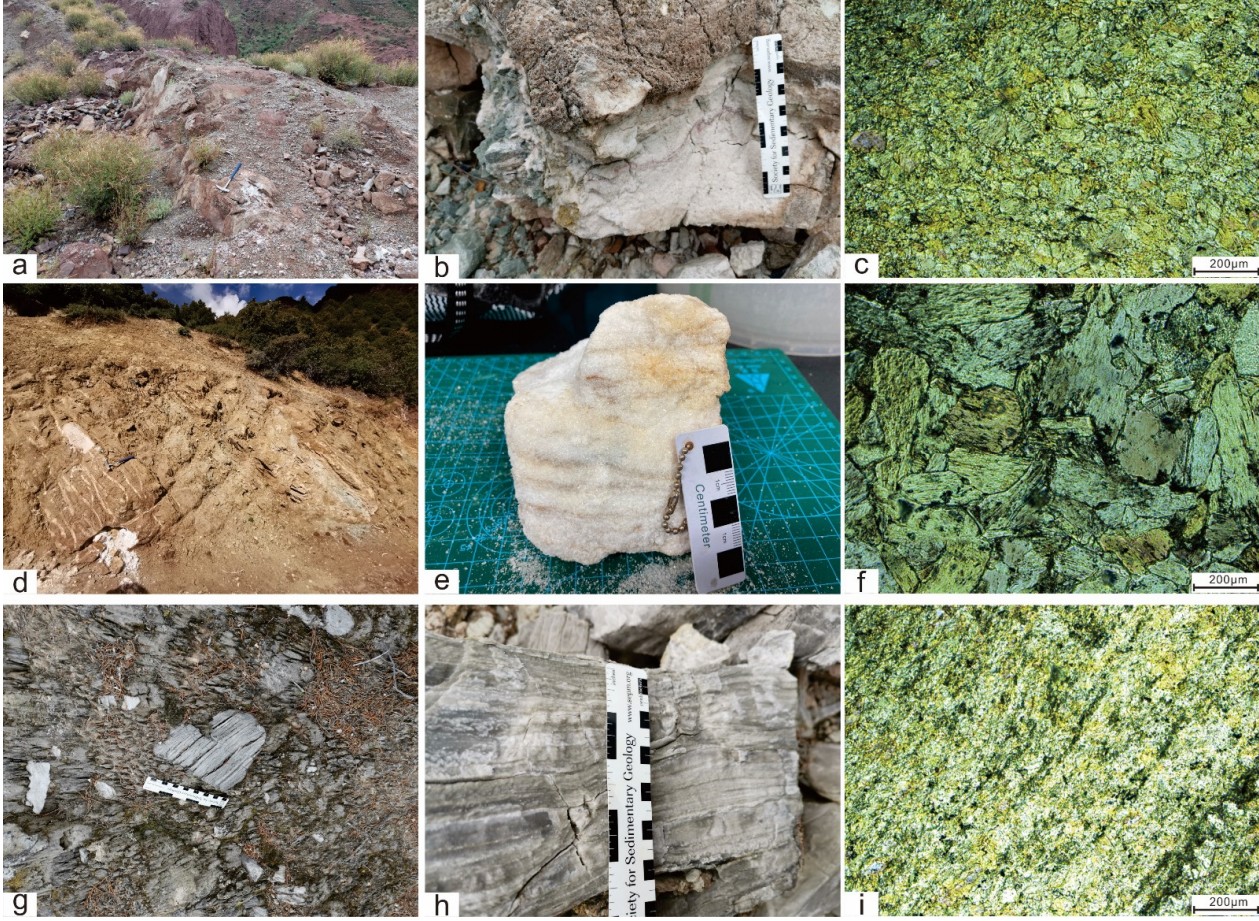

**Figure 4.** Characteristics of the Middle Jurassic evaporites in the Qamdo Basin. (**a**) lenticular gypsum interbedded with purplish sandstone, Rumei; (**b**) unconsolidated gypsum mixed with carbonate breccia, Rumei; (**c**) subhedral, amoeboid gypsum crystals, Rumei, cross-polarised light (CPL); (**d**) bedded gypsum intercalated with brown siltsone, Rigei; (**e**) centimetre-scale gypsum layers characterised by the number of detrital components, Rigei, CPL; (**f**) subhedral to euhedral gypsum crystals, ~100–500 μm, Rigei; (**g**) millimetre-scale gypsum layers with corroded surfaces, Huola; (**h**) gypsum laminae, Huola; and (**i**) aligned gypsum fibres, Huola, CPL.

The scanning electron microscopy (SEM) (Thermo Fisher, Waltham, MA, USA) analysis was carried out at the Key Laboratory of Deep-Earth Dynamics, Institute of Geology, using an FEI Nova NanoSEM 450. The backscattered electron (BSE) images were taken under an operating voltage of 15–20 KV and a working distance of 13.5 mm. The Sr isotope analyses were performed at the Beijing Research Institute of Uranium Geology. About 10 mg of sample (200 mesh) were dissolved using 4 M $HNO_3$ after being washed with milli-Q water and dried. The Sr was extracted from the Sr-resin and was analysed using a Neptune Plus multicollector inductively coupled plasma mass spectrometer (ICP-MS). The Sr isotope data are reported as $^{87}Sr/^{86}Sr$. $^{87}Sr/^{86}Sr$ values of 0.71022 to 0.71030 were obtained for standard NBS987. The reported uncertainties of the $^{87}Sr/^{86}Sr$ ratios are 1σ. For the S isotope analysis, the samples were purified as $BaSO_4$ after combustion with an Eschka reagent, and then, they were treated with $V_2O_5$ to produce $SO_2$. The resulting $SO_2$ was measured using the MAT 251 EM mass spectrometer (Thermo Fisher, Waltham, MA, USA) at the MNR Key Laboratory of Metallogeny and Mineral Assessment, Beijing. The $\delta^{34}S$ values are reported relative to Canyon Diablo troilite (CDT), and the estimated error is about ±0.2‰. The detailed procedure used for the ICP-MS trace element analysis has been described by [22].

## 4. Results

The $^{87}Sr/^{86}Sr$ ratios of all of the gypsum samples from Huola, Rigei, and Rumei range from 0.707531 to 0.708163, which are much higher than those of Middle Jurassic seawater (ca. 0.70732 to 0.70684 [23]). The $^{87}Sr/^{86}Sr$ ratios of the Huola samples fall within a narrow range of 0.70602 to 0.707729. The Rigei and Rumei samples have a more scattered range of $^{87}Sr/^{86}Sr$ ratios: 0.707531 to 0.707886 and 0.707602 to 0.708163, respectively (Table 2).

**Table 2.** Sr and S isotope compositions and trace elements of sulphate samples from Middle Jurassic Dongdaqiao Formation, Qamdo Basin.

| Sample ID | Location | $^{87}Sr/^{86}Sr$ | $\Delta^{34}S_{V-CDT}$ | Rb | Sr | Mg | Na | Si | Ca | Rb/Sr |
| --- | --- | --- | --- | --- | --- | --- | --- | --- | --- | --- |
| | | | | (ppm) | (ppm) | (ppm) | (ppm) | (%) | (%) | |
| MKHL-20 | | 0.707729 ± 0.000017 | 16.3 | 1.71 | 483 | 4206 | 680 | 1.01 | 32.06 | 0.0035 |
| MKHL-21 | | 0.707628 ± 0.000011 | 15.8 | 2.61 | 1405 | 2874 | 1150 | 1.67 | 31.16 | 0.0019 |
| MKHL-22 | | 0.707668 ± 0.000018 | 15.5 | 1.4 | 1048 | 1572 | 460 | 0.993 | 31.97 | 0.0013 |
| MKHL-23 | Huola | 0.707659 ± 0.000016 | 15.8 | 2.7 | 2132 | 4806 | 1160 | 2.08 | 31.12 | 0.0013 |
| MKHL-24 | | 0.707683 ± 0.000011 | 16.1 | 0.585 | 855 | 768 | 500 | 0.378 | 32.17 | 0.0007 |
| MKHL-25 | | 0.707602 ± 0.000013 | 15.5 | 0.867 | 1236 | 1920 | 1490 | 0.746 | 31.8 | 0.0007 |
| MKHL-26 | | 0.707608 ± 0.000014 | 15.8 | 1.6 | 1206 | 2940 | 740 | 1.1 | 31.65 | 0.0013 |
| MKHL-27 | | 0.707608 ± 0.000012 | 15.9 | 1.48 | 1002 | 2052 | 440 | 0.603 | 31.84 | 0.0015 |
| MKRG-28 | | 0.707531 ± 0.000013 | 16 | 0.267 | 1057 | 2532 | 450 | 0.184 | 32.14 | 0.0003 |
| MKRG-29 | Rigei | 0.707715 ± 0.000018 | 20.0 | 2.78 | 2084 | 4356 | 540 | 0.897 | 31.66 | 0.0013 |
| MKRG-30 | | 0.707661 ± 0.000017 | 20.5 | 0.11 | 2281 | 378 | 400 | <0.010 | 32.32 | 0.0000 |
| MKRG-31 | | 0.707886 ± 0.000020 | 20.8 | 0.069 | 2521 | 342 | 510 | <0.010 | 32.48 | 0.0000 |
| MKRM-33 | | 0.707854 ± 0.000020 | 15.5 | 0.258 | 1211 | 774 | 360 | 1.18 | 31.8 | 0.0002 |
| MKRM-34 | Rumei | 0.707627 ± 0.000012 | 16.1 | 0.508 | 943 | 738 | 570 | 0.385 | 32.31 | 0.0005 |
| MKRM-35 | | 0.708163 ± 0.000011 | 15.3 | 1.29 | 1190 | 2250 | 810 | 2.27 | 33.51 | 0.0011 |
| MKRM-36 | | 0.707602 ± 0.000008 | 16.3 | 0.71 | 890 | 2556 | 490 | 0.472 | 32.05 | 0.0008 |

The $\delta^{34}S$ values of the Huola and Rumei samples are relatively consistent, ranging from 15.3‰ to 16.3‰ (Huola: 15.5‰ to 16.3‰, mean value = 15.8‰; Rumei: 15.3‰ to 16.3‰, mean value = 15.8‰). The Rigei samples have more variable values, with overall higher $\delta^{34}S$ values (16‰, 20‰, 20.5‰, and 20.8‰ for four samples) (Table 2).

The Sr, Mg, and Na contents of Huola samples are 483–2132, 768–4806, and 440–1490 ppm; those of the Rigei samples are 1057–2521, 342–4356, and 400–540 ppm; and those of the Rumei Samples are 890–1211, 738–2556, and 360–810 ppm (Table 2). Figure 5 shows that the Sr and Na contents, Sr and Mg contents, and Na and Mg contents of the gypsum samples are not well correlated. The Mg and Na concentrations of some of the Rigei samples are significantly lower

than their Sr concentrations compared to the other samples (Figure 5), which correspond to higher $\delta^{34}S$ values (Table 2).

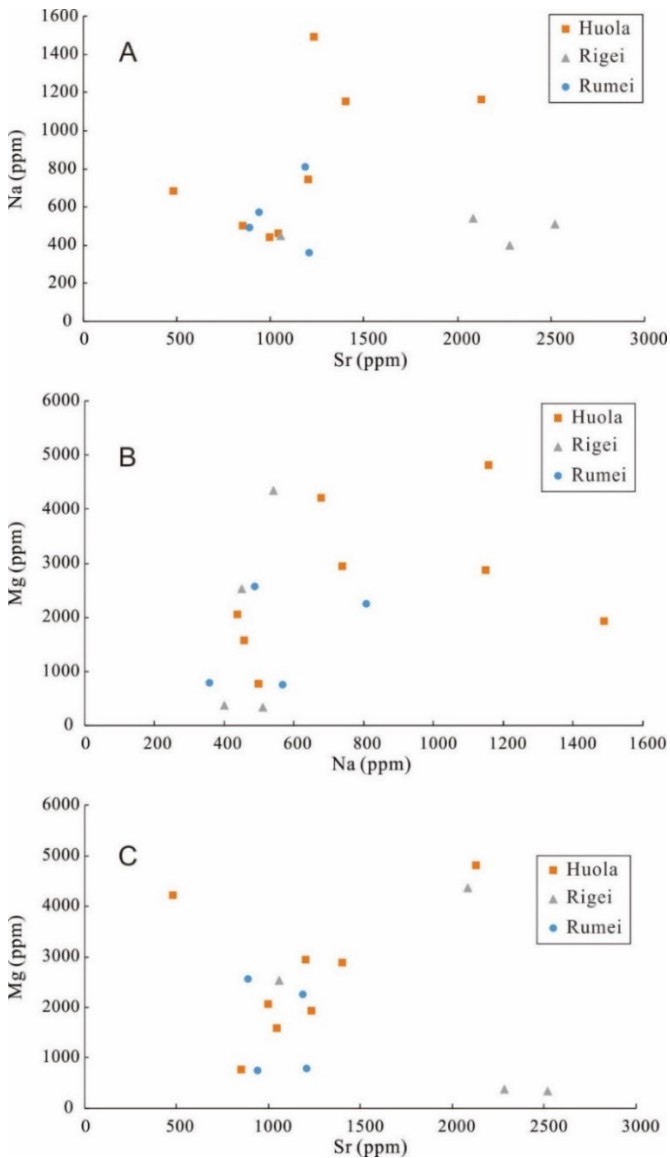

**Figure 5.** Sr vs. Na (**A**), Na vs. Mg (**B**), and Sr vs. Mg (**C**) of gypsum samples from Huola, Rigei, and Rumei in the Qamdo Basin.

## 5. Discussion

### 5.1. Sr Isotopes

Before the origin of the brines from which these sulphates precipitated could be analysed based on their Sr isotope compositions, it was necessary to evaluate the other sources of Sr, in addition to the major contributors (seawater and non-marine fluids). Detrital clay and feldspar, which usually contain radiogenic $^{87}Sr$, only account for a small portion of the Sr because the Si contents of most of the samples are less than 1% (Table 2). The low Sr contents (100 ppm, [24]) and relatively robust crystal structures of the clay minerals imply that the detrital minerals had a negligible effect on the Sr isotope composition of the surrounding brine. Denison et al. (1998) [25] demonstrated that impermeable gypsum and anhydrite could resist interaction with extraneous fluids, and resorption of post-depositional waters by re-hydration of anhydrite to gypsum is unlikely to cause a notable change in the $^{87}Sr/^{86}Sr$ ratio. The influence of radiogenic $^{87}Sr$ on the isotope compositions of the sulphate

samples was also determined to be negligible based on the extremely low Rb/Sr ratios of the samples (Table 2). Thus, the Sr isotope compositions of the evaporites within the basin reflect the composition of the parent brines derived from the major potential sources.

The Sr isotopic composition of seawater fluctuated during the Phanerozoic [26]. The general trend of the [87]Sr/[86]Sr variations of Phanerozoic seawater has been determined in previous studies [11,23,26]. The strontium isotope curve for Middle Jurassic seawater is characterised by the lowest [87]Sr/[86]Sr ratios and a decreasing trend (i.e., monotonically decreasing from 0.70732 to 0.70684 from the Aalenian to the Callovian). The [87]Sr/[86]Sr ratios of all of the samples from the Middle Jurassic Dongdaqiao Formation are higher than the range of contemporaneous seawater and are widely scattered (Figure 6), indicating that continental water with more radiogenic [87]Sr was added to the evaporite basin [27]. If the parameters (Sr concentration and isotopic composition) of the different sources (mainly seawater and continental water in this study) that controlled the [87]Sr/[86]Sr ratios of the parent brines can be defined, it is possible to quantitatively evaluate the contributions of these sources.

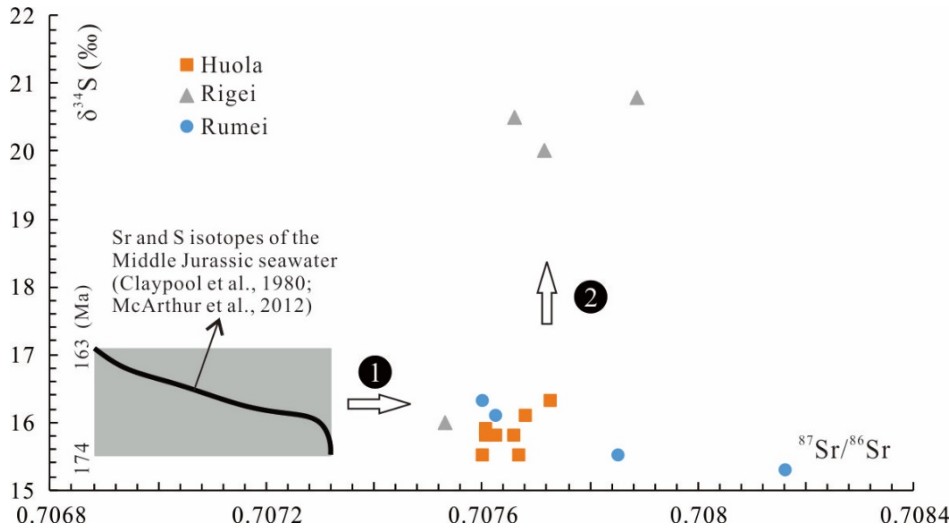

**Figure 6.** Sr vs. S isotopes of Middle Jurassic sulphates in the Qamdo Basin. Process 1: the inflow of continental flux with more radiogenic [87]Sr elevated the [87]Sr/[86]Sr ratios of evaporates; process 2: bacterial sulphate reduction increased the $\delta^{34}$S values [9,23].

Although there is no direct evidence supporting the hypothesis that Middle Jurassic seawater had the same Sr concentration as modern seawater (8 ppm), Kirkland et al. (1995) [27] concluded that it is very unlikely that the Sr concentration was outside the range of 6 to 12 ppm. The accurate formation ages of the gypsum samples are not available. The evaporitic intercalations within the Dongdaqiao sedimentary sequence only indicate that these gypsum samples were formed during the Middle Jurassic, thus defining a seawater [87]Sr/[86]Sr ratios range of ca. 0.70684 to 0.70732 [23]. There is no method of obtaining the Sr concentration and isotopic compositions of Middle Jurassic river water. However, the Sr concentration and isotopic compositions of the modern river adjacent to the evaporites are well constrained. The Sr concentration of the Lancangjiang River ranges from 0.30 to 0.66 ppm, and its [87]Sr/[86]Sr ratios range from 0.70974 to 0.71020 [28]. We used a two-component mixing model and the following equation:

$$^{87}Sr/^{86}Sr_m = \,^{87}Sr/^{86}Sr_s \times C_s + \,^{87}Sr/^{86}Sr_r \times C_r$$

$^{87}Sr/^{86}Sr_m$, $^{87}Sr/^{86}Sr_s$, and $^{87}Sr/^{86}Sr_r$ are the Sr isotope compositions of the mixed parent brine, Middle Jurassic seawater, and Middle Jurassic river water, respectively. $C_s$ and $C_r$ are the proportions of the Sr contributions from seawater and river water, respectively,

and $C_s + C_r = 1$. The largest and smallest proportions of Sr supplied by the river water were 45.6% and 11.6%, respectively, based on our calculations (Table 3).

**Table 3.** The model calculation showing the contributions of marine and continental waters.

| Sr Contribution of River Water (%) | Parameters | | |
|---|---|---|---|
| | $^{87}Sr/^{86}Sr$ of River Water | $^{87}Sr/^{86}Sr$ of Sea Water | $^{87}Sr/^{86}Sr$ of Evaporites |
| 45.6% | 0.70974 (minimum) | 0.70684 (minimum) | 0.708163 (maximum) |
| 11.6% | 0.7102 (maximum) | 0.70732 (maximum) | 0.707602 (minimum) |

According to the assumptions regarding the Sr concentrations of the river water and seawater, the river/sea water ratios with respect to the water volume were calculated to be 1 to 33. Thus, it is suggested that the parent brines from which the sulphates precipitated were mainly derived from continental water with respect to water volume.

*5.2. S Isotopes*

Most of the $\delta^{34}S$ values (+15.3 to +16.3‰) plot within the range of contemporaneous seawater (Figure 6) (+16.3 ± 0.8‰ [8]), except for three samples from Rigei, which have $\delta^{34}S$ values of +20 to +20.8‰. These abnormally high $\delta^{34}S$ values were likely caused by bacterial sulphate reduction (BSR) because BSR preferentially removes $^{32}S$ from the dissolved sulphate and leads to the $SO_4^{2-}$ in the residual brine being enriched in $^{34}S$ [29]. In addition to these samples with anomalously high $\delta^{34}S$ values, the very limited variation in the $\delta^{34}S$ values (+15.3 to +16.3‰) could have been caused by sulphur isotope fractionation, albeit an insignificant amount [30]. Seawater evaporation experiments have demonstrated that during the precipitation of gypsum, the $\delta^{34}S$ values of the precipitates formed during the final stage are depleted by ~1‰ relative to the precipitates formed during the initial stage (+21‰ vs. +22‰ [31]). The offset of our $\delta^{34}S$ values (~1‰) is consistent with that caused by fractionation during seawater evaporation.

Alternatively, the slightly lower $\delta^{34}S$ values could have originated from the input of $^{34}S$-depleted continental water. If this is the case, it is analogous to the calculation of marine and non-marine contributions based on Sr isotopes.

Modern seawater has an $SO_4^{2-}$ concentration of ~2800 ppm [25]. Based on analyses of primary halite fluid inclusions, the $SO_4^{2-}$ concentration of Phanerozoic seawater varied significantly [32]. We postulate that the Middle Jurassic seawater, which flowed into the Qamdo salina, had a similar $SO_4^{2-}$ concentration to that of Late Jurassic seawater, i.e., ~670 to 1340 ppm (median = 1005 ppm) [32]. Modern rivers have a mean $SO_4^{2-}$ concentration of ~11 ppm [25], and the $\delta^{34}S$ of river water is typically thought to be between 5‰ and 15‰ [33–35]. A recent study revealed that the $\delta^{34}S$ value of modern riverine sulphate is 4.4 ± 4.5‰ based on measurements of rivers draining different geographical and climate regions [36]. The $SO_4^{2-}$ concentration of the Changjiang River ranges from 3.2 to 28.9 mg/L [37]. Thus, we assumed that the riverine water that flowed into the Qamdo Basin probably had an $SO_4^{2-}$ concentration of 3 to 30 ppm and $\delta^{34}S$ values of 0 to 15‰. Using various parameters (i.e., the sulphate concentrations and isotope compositions of seawater and river water), mixing curves were calculated, and it was found that seawater can be very sensitive to continental contributions when the river water has relatively high $SO_4^{2-}$ concentrations and low $\delta^{34}S$ values (Figure 7).

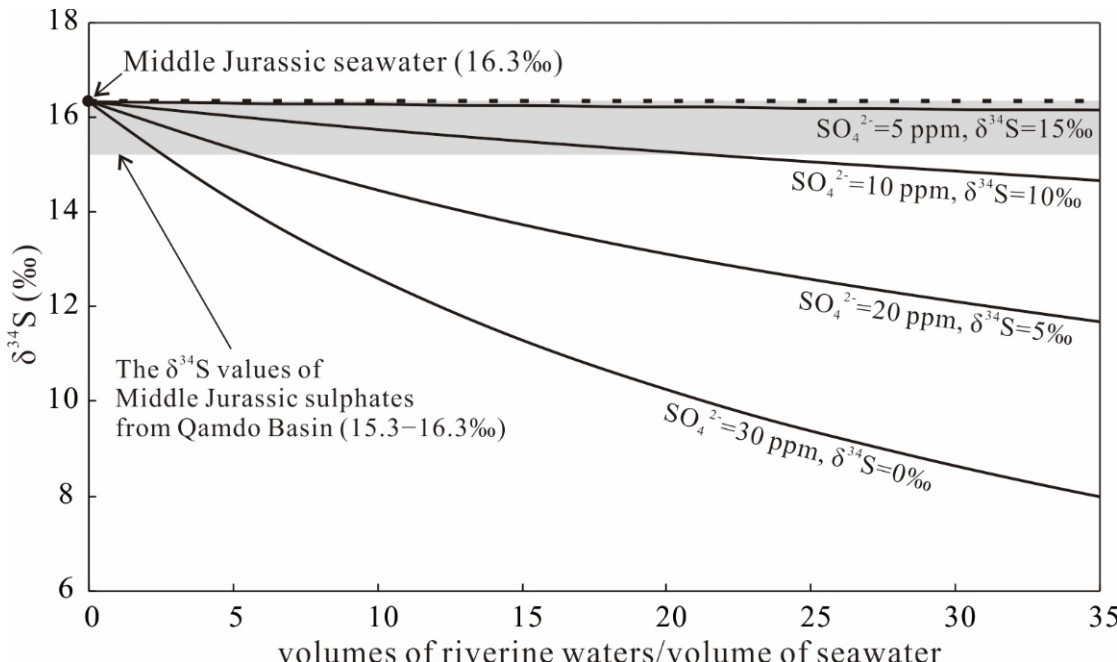

**Figure 7.** Mixing curves for various $SO_4^{2-}$ concentrations and $\delta^{34}S$ values of riverine waters with seawater ($\delta^{34}S$ = 16.3‰, $SO_4^{2-}$ concentration = 1005 ppm).

According to the calculation results based on the Sr mixing equation, 1 to 33 volume units of river waters per volume unit of seawater were required to cause the corresponding Sr isotope shift. A comprehensive study of the sulphur isotope composition of global river water has shown that the $\delta^{34}S$ values of river water seldom exceed 10‰ [36]. If we use this isotopic value (+10‰), the $SO_4^{2-}$ concentrations of the river water should not exceed 10 ppm (Figure 7), which is consistent with the results (11 ppm) of Denison et al., (1998) [25]. However, the statistical results for various rivers have shown that over half of the rivers studied had $SO_4^{2-}$ concentrations of greater than 11 ppm (mean value = 32 ppm [36]). Thus, we assumed that the mean $\delta^{34}S$ value of the river water supplied to the Qamdo Basin was probably greater than 10‰. In our study area, the gypsum sequences were also deposited during the Early Triassic in the immediate vicinity of the locations of the Middle Jurassic gypsums, i.e., Xiaochangdu and Yanjing (Figure 2A). The $\delta^{34}S$ values of these Late Triassic gypsums range from 15.4‰ to 16.6‰ (unpublished data), which are consistent with values for contemporaneous open marine environments [38]. In the meantime, the Triassic evaporites have $^{87}Sr/^{86}Sr$ ratios ranging from ca. 0.7074 to 0.7080 [23], which are slightly higher than those of Jurassic seawater. Thus, the scenario in which runoff dissolved the Late Triassic gypsum and then drained into the Qamdo Basin during the Middle Jurassic would produce nearly no shift in the S isotope values but enhance the $^{87}Sr/^{86}Sr$ ratios of mixed parent brines. If this was the case, the $SO_4^{2-}$ concentration of the river water would be essentially immaterial.

*5.3. Trace Elements*

Lu et al. (1997) [39] demonstrated that the Na and Cl in fluid and solid inclusions account for a large proportion of the total Na and Cl contents of gypsum and anhydrite, whereas the Mg and Sr contents of gypsum and anhydrite are the least affected by the composition of the inclusions. This is because Cl does not occur in the lattice of gypsum and anhydrite [39,40], and $Na^+$ and $Ca^{2+}$ have different valence states, which make inclusions the predominant source of Cl and Na. However, Sr can easily substitute for Ca during the precipitation of gypsum and anhydrite, and the Sr contents of inclusions are very low. Mg and Ca have the same valence state (2+). Thus, Sr and Mg may not primarily reside in fluid or solid inclusions, and they mostly exist in the gypsum and anhydrite lattice where they

substitute for Ca. The Mg and Sr concentrations of many of the samples experienced very little variation after whole sample dissolution, grinding, decrepitation, and rinsing with ethyl alcohol [39]. It has been argued that Mg and Sr mainly substitute for Ca in the lattice of gypsum as well [39]. Therefore, the Mg and Sr contents of gypsum are mainly controlled by the salinity of the brine, that is, under the same conditions, the higher the salinity of the brine is, the higher the Mg and Sr contents of the gypsum are. In this study, we only cleaned the surface of the samples without further treatment (grinding, decrepitation, and rinsing with ethyl alcohol as proposed by [39]). Thus, only the Sr and Mg contents are utilised in the following discussion.

For the Huola samples, the Mg and Sr contents are well correlated, except for one outlier ($R^2$ = 0.89; Figure 5C). This suggests that the Mg and Sr contents increase with increasing salinity. It is possible that the data points for the Rumei samples could exhibit a linear relationship if more samples were analysed since these data points are relatively convergent and are located on the trend formed by the Huola samples (Figure 5C). In contrast, the Rigei samples are more scattered compared to those from Huola and Rumei, especially two points that plot beyond the main trend (extremely high Sr and low Mg contents, Figure 5C).

The higher $\delta^{34}S$ values of the gypsum samples from Rigei suggest that BSR was prevalent. The Sr contents of the samples with higher $\delta^{34}S$ values are also much higher (Sr = 2084–2521 ppm) than those of the other samples (Sr = 483–1405 ppm, with the exception of 2132 ppm) (Figure 8). The SEM analysis revealed that some celestite ($SrSO_4$) crystals occur in these samples (Figure 9). Thus, the elevated Sr contents of the high $\delta^{34}S$ samples contributed to the formation of celestite. The relatively low solubility of celestite indicates that it is difficult to transport significant amounts of Sr and $SO_4$ ions [29]. Therefore, the Sr and $SO_4$ ions from which the celestite formed were likely provided by an in situ source. BSR consumed the $SO_4$, which increased the Sr concentration of the brine, thus promoting the precipitation of celestite.

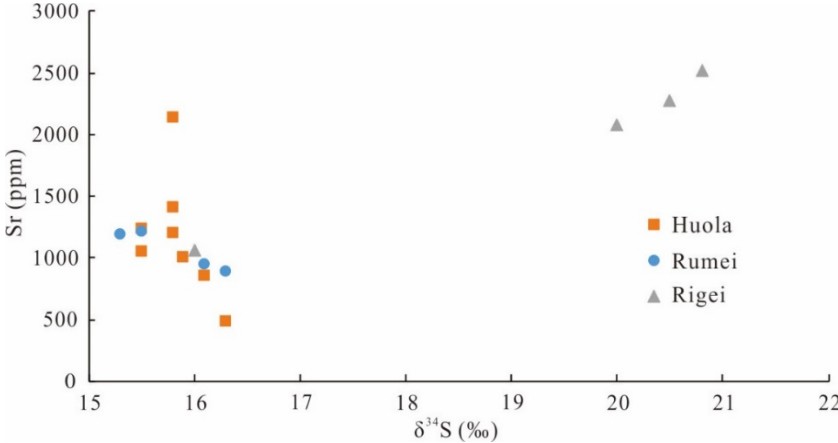

**Figure 8.** The Sr contents and S isotope compositions of gypsums from the Qamdo Basin.

The linear correlation between the Mg and Sr contents of the samples from Huola indicates that as was discussed above, these gypsum samples formed during an increasingly more saline environment. Except for three samples with abnormally high Sr contents, the Sr contents and S isotope values are roughly negatively correlated (Figure 8), which suggests that the gypsum precipitation became gradually more depleted in $^{34}S$ as the evaporation progressed. As a result, the variations in the $\delta^{34}S$ values (15.3‰ to 16.3‰) of the samples that were not affected by BSR were likely caused by S isotope fractionation during evaporation.

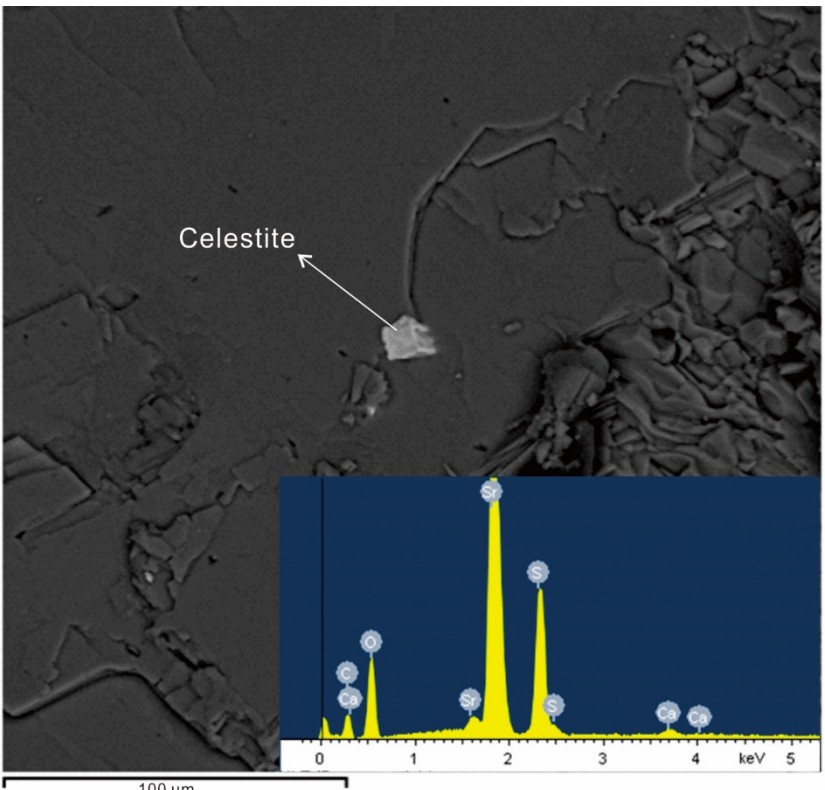

**Figure 9.** SEM image of gypsum sample from Rigei showing the occurrence of celestite.

In summary, the abnormally high Sr contents of the gypsum samples are attributed to the formation of celestite facilitated by BSR. The Mg and Sr contents can be used as indicators of the extent of the evaporation if no other processes exist other than evaporation. The trace element contents corroborate the conclusion that the variations in the S isotope compositions were mainly controlled by the reservoir effect during the evaporation and BSR.

*5.4. Provenance and Palaeogeographic Implications*

The Sr isotope compositions of the Middle Jurassic gypsum samples from the Qamdo Basin imply that water with elevated $^{87}Sr/^{86}Sr$ ratios mixed with Jurassic seawater. The model calculations indicate that the river/sea water ratio with respect to volume was 1 to 33. This indicates that continental water prevailed over seawater. However, the S isotopes indicate that the continental water exerted little influence on the integrated S isotope signature. The recycling of Triassic evaporites may have obscured the continental S isotope signatures (i.e., characterised by low $\delta^{34}S$ values). The Sr and S isotope systematics of the gypsum samples suggest that the S isotope compositions are more consistent with a marine signature, while the Sr isotopes are more scattered. This feature resembles those of the evaporites in the Jurassic Todilto Formation in New Mexico, the Permian Blaine Formation in Blaine County, Oklahoma, the Permian Salado Formation in New Mexico [25], and the Cretaceous evaporites in the Lanping-Simao Basin [41]. The common features of these evaporites indicate that S isotopes are less sensitive to meteoric influences than Sr isotopes, and S isotopes are useful in identifying waterbodies with a marine base, while Sr isotopes can be used to recognise and quantify substantial continental contributions [20]. The reason for such a discrepancy may be due to the relatively large offset between river water and seawater with respect to Sr isotopes compared to that of S isotopes as discussed in 5.2. From this point of view, the Sr isotopes of evaporites are more reliable to identify the provenance of evaporites. Therefore, we suggest that the Middle Jurassic evaporites in the Qamdo Basin were derived from the overflow of seawater mixed with a relatively large

amount of continental water. The occurrence of both marine and non-marine fossils within the Dongdaqiao Formation [19] also indicates that marine and non-marine water sources recharged the Qamdo Basin.

As mentioned in the geologic background, the lithology of the Dongdaqiao Formation (bedded fine sandstones, conglomerates, thinly bedded shales, and mudstones with sparsely distributed evaporites) in the Qamdo Basin indicates a paralic environment with a large amount of clastic input [19], whereas, in the Qiangtang Basin, sandstones and mudstones with horizontal bedding, parallel bedding, and cross-bedding of Quemocuo Formation indicating deltaic and tidal flat environments, and carbonates with horizontal bedding of Buqu Formation indicating neritic environment. The comparison of the Middle Jurassic lithologies in the Qamdo and Qiangtang basins (Figure 3) suggests that the Qiangtang Basin was more affected by the marine transgression than the Qamdo Basin.

Currently, no published Sr isotope data for the anhydrites in the Qiangtang Basin are available, thus it is not feasible to compare the Sr isotope compositions of the sulphates in the Qamdo and Qiangtang basins. However, the Sr isotope compositions of the Middle Jurassic marine carbonates in the Qiangtang Basin have been analysed [42]. The $^{87}Sr/^{86}Sr$ ratios of the carbonates in the Qiangtang Basin range from 0.70702 to 0.70747 [42], which are consistent with those of Middle Jurassic seawater (Figure 10). This suggests that in the Qiangtang Basin, the water body from which the evaporites precipitated was mainly derived from seawater, whereas the sulphates in Qamdo Basin have $^{87}Sr/^{86}Sr$ ratios higher than those of the carbonates in the Qiangtang Basin and Middle Jurassic seawater (Figure 10). Thus, the discrepancy between the Qiangtang and Qamdo basins with respect to the Sr isotope compositions is also consistent with the fact that the evaporites in the Qiangtang Basin exhibit stronger marine characteristics than those in the Qamdo Basin.

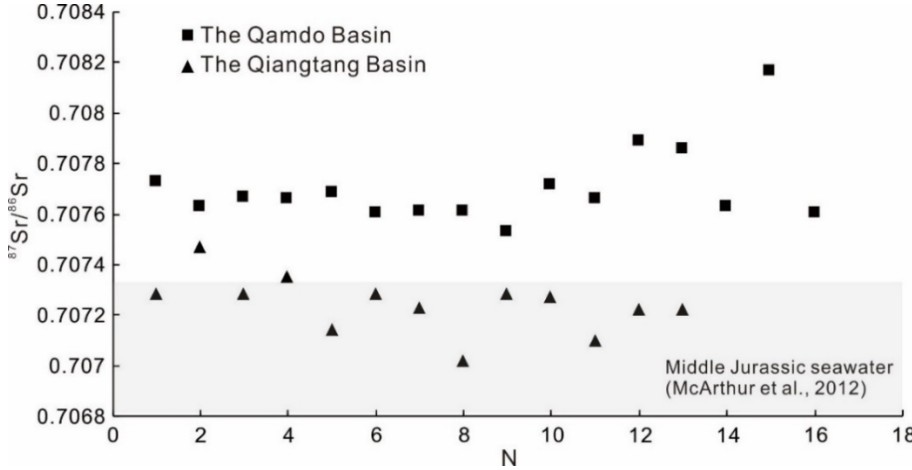

**Figure 10.** The comparison among the Sr isotope compositions of sulphates in Qamdo Basin, carbonates in Qiangtang Basin, and Middle Jurassic seawater [23].

Consequently, the lithological correlation and Sr isotope comparison of the Qiangtang and Qamdo basins indicate that the Qiangtang Basin was connected to the open sea, and the brine in the Qiangtang Basin may have overflowed into the Qamdo Basin intermittently (Figure 11). During the Middle Jurassic, the Qiangtang Basin was connected to the open sea to the southwest, and a suite of marine strata consisting of carbonates and evaporites was formed during the transgression. The distal Qamdo Basin received a clastic influx from the emerged lands, engendering a major accumulation of clastic sequences. The brine in the Qiangtang Basin with a marine signature flowed into the Qamdo Basin, resulting in a marine base for the precipitation of evaporites. The seawater may have evaporated to some extent before flowing into the Qamdo Basin, which is probably the reason why the Dongdaqiao Formation is lacking carbonates. This palaeogeographic configuration is similar to that of the Middle Triassic to the Early Jurassic evaporite of Iberia [43]. In

the eastern Iberian platform, clastic sediments had been formed with nodular anhydrite proximal to stable meseta. At the same time, a carbonate-evaporite platform was formed in the shallow epicontinental sea, distal to the highland [43]. The analogy between evaporites from Middle Jurassic Qiangtang-Qamdo Basins and from the Early Mesozoic Iberian platform corroborates that the parent brines in the Qamdo Basin were likely derived from the Qiangtang Basin which was connected to the open sea. Nevertheless, we cannot preclude the occurrence of a scenario in which the open sea discharged directly into the Qamdo Basin from another seaway(s). Regardless of which scenario occurred, the quantity of seawater-derived fluids was small during the sedimentation of the Middle Jurassic strata in the Qamdo Basin.

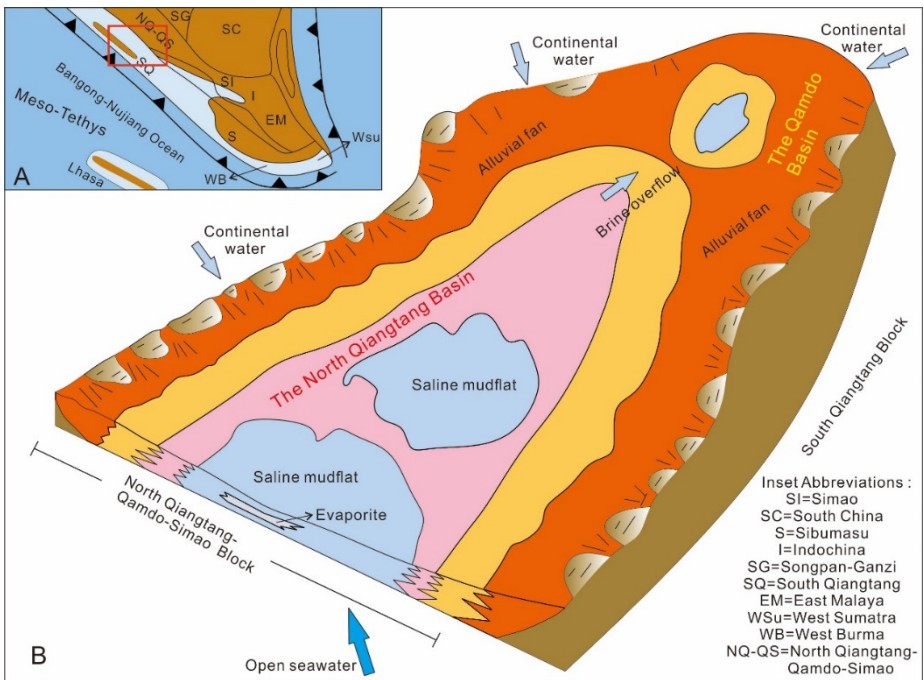

**Figure 11.** Palaeogeographic model of the Middle Jurassic sedimentation in Qamdo Basin and adjacent Qaingtang Basin. (**A**), Palaeogeographic reconstruction for eastern Tethys (Metcalfe, 2013); (**B**), Discharge model for Qamdo and Qiangtang Basins.

## 6. Conclusions

(1) The range of the $^{87}$Sr/$^{86}$Sr ratios (0.707602 to 0.708163) of the Middle Jurassic gypsum in the Qamdo Basin is higher than that of contemporaneous seawater, which indicates that the Middle Jurassic evaporites in the Qamdo Basin were likely derived from continental water mixed with seawater. Moreover, based on the model calculations, the continental water prevailed over the seawater.

(2) The majority of the Middle Jurassic gypsum samples from the Qamdo Basin have $\delta^{34}$S values of 15.3‰ to 16.3‰, which are consistent with those of contemporaneous seawater, suggesting that continental water exerted a negligible influence on the SO$_4$. This marine signature could have been affected by Triassic evaporite recycling with similar S isotope compositions. The elevated $\delta^{34}$S values of three samples were caused by bacterial sulphate reduction, which is corroborated by their trace element compositions and SEM observations (the occurrence of celestite).

(3) The lithological correlation and isotope comparison of the Qiangtang and Qamdo Basins indicate that the Qiangtang Basin was connected to the open sea, and the brine in the Qiangtang Basin may have overflowed into the Qamdo Basin intermittently. Further investigation is needed to substantiate the conclusions of this study due to the low number of samples from the Middle Jurassic sequence.

**Author Contributions:** Conceptualization, L.S. and J.F.; methodology, J.F.; validation, J.F., L.S., X.G. and Z.S.; investigation, L.S., J.F., X.G. and Z.S.; data curation, L.S.; writing—original draft preparation, J.F.; writing—review and editing, L.S. All authors have read and agreed to the published version of the manuscript.

**Funding:** This research was funded by the national Key Project for Basic Research of China (No. 2011CB403007).

**Data Availability Statement:** Not applicable.

**Conflicts of Interest:** The authors declare no conflict of interest.

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
