# Peer review of "S and Sr Isotope Compositions and Trace Element Compositions of the Middle Jurassic Evaporites in Eastern Tibet: Provenance and Palaeogeographic Implications"

_minerals, doi:10.3390/min12081039_

Round 1

Reviewer 1 Report

- The authors should add in table 1 a brief petrographic description of each of the analyzed samples

- Review the photomicrographs of fig 2 and its corresponding figure captions . The scale is not well observable.

-The conclusions must be established as provisional due to the low number of samples studied.

Author Response

 -The authors should add in table 1 a brief petrographic description of each of the analyzed samples

Response: Table 1 with lithology descriptions has been added to the manuscript

- Review the photomicrographs of fig 2 and its corresponding figure captions . The scale is not well observable.

Response: The font of scale's been amplified, and the line is thickened, thus we did not annotate the scale in the caption

-The conclusions must be established as provisional due to the low number of samples studied.

Response: The last sentence of conclusion points out that further investigation is needed due to low number of samples.

Reviewer 2 Report

The manuscript titled "S and Sr isotope composition and trace element compositions of the Middle Jurassic evaporates in eastern Tibet ..." by Jinna Fei and others describes the origin of evaporates in the Qamdo Basin of eastern Tibet and offers interpretations of their provenance based on their S and Sr isotopic compositions.  Interpretations based on the Sulphur isotopic data seem to contradict the Strontium isotopic data and the resolution of these seemingly contradicting data is attempted although done so in a less than straightforward way resulting in unnecessary confusion.  The authors conclude that the Sr isotope data indicates that the analyzed evaporates "were derived from continental water mixed with seawater" and that "continental water prevailed over seawater". However, they find that the S isotopic data suggests that "continental water exerted a negligible influence" on the majority of the analyzed gypsum samples and that they "are consistent with those of contemporaneous seawater".   This contradiction is briefly discussed but a clear explanation is not provided.  Finally, it is concluded that the 87Sr enriched continental waters of the Qamdo basin may have been influence by overflow of seawater from the adjacent Qiangtang Basin that was connected to the open sea.  This final conclusion is ambiguous. The authors try to have it both ways, but which is more reliable; the S data or the Sr data? and why?  The water can't be both "consistent with seawater" and "continental water that prevailed over seawater". Which is it?   It may simply involve the clarification of a few concepts that may seem obvious to the authors, but some important clarification is needed.

Line 20. Replace "has" with "samples have"

Figure 1. The map should have the Dongdaqiao Formation on it.

Line 92. Replace Seventeen with Sixteen (based on Table 1). 

Line 145. The positive correlation is not clear.  Delete "In addition to a few spots"

Line 169. Delete the first "of".

Line 284. Delete "of"

Figure 6. This figure is redundant. Delete it or delete Figure 3c.

Line 309. Replace "in addition to" with "Except for"

Line 311. Replace "precipitated" with "precipitation"

Author Response

The manuscript titled "S and Sr isotope composition and trace element compositions of the Middle Jurassic evaporates in eastern Tibet ..." by Jinna Fei and others describes the origin of evaporates in the Qamdo Basin of eastern Tibet and offers interpretations of their provenance based on their S and Sr isotopic compositions.  Interpretations based on the Sulphur isotopic data seem to contradict the Strontium isotopic data and the resolution of these seemingly contradicting data is attempted although done so in a less than straightforward way resulting in unnecessary confusion.  The authors conclude that the Sr isotope data indicates that the analyzed evaporates "were derived from continental water mixed with seawater" and that "continental water prevailed over seawater". However, they find that the S isotopic data suggests that "continental water exerted a negligible influence" on the majority of the analyzed gypsum samples and that they "are consistent with those of contemporaneous seawater".   This contradiction is briefly discussed but a clear explanation is not provided.  Finally, it is concluded that the 87Sr enriched continental waters of the Qamdo basin may have been influence by overflow of seawater from the adjacent Qiangtang Basin that was connected to the open sea.  This final conclusion is ambiguous. The authors try to have it both ways, but which is more reliable; the S data or the Sr data? and why?  The water can't be both "consistent with seawater" and "continental water that prevailed over seawater". Which is it?   It may simply involve the clarification of a few concepts that may seem obvious to the authors, but some important clarification is needed.

Response: This question is very meaningful and helpful to improve this manuscript. The reason for such a discrepancy resulting from Sr and S isotopes is discussed in the text: several evaporite deposits have shown similar Sr vs. S patterns (in 5.2), which suggest Sr isotopes of marine signature are more sensitive to continental river waters than S isotopes. From this point of view, we suggest that Sr isotopes are preferred to be used to identify the origin of the parent brines in which evaporite precipitated. The reason for such a discrepancy is that the recycling of Triassic gypsums in the vicinity of Jurassic evaporites in the Qamdo Basin have nearly identical S isotopes and elevated 87Sr/86Sr ratios compared to those of Jurassic seawater, which could produce such a Sr vs. S isotopic systematic of evaporites.

Line 20. Replace "has" with "samples have"

Response: already replaced

Figure 1. The map should have the Dongdaqiao Formation on it.

The Dongdaqiao Formation has been outlined and added to Figure 2.

Line 92. Replace Seventeen with Sixteen (based on Table 1). 

Response: already replaced

Line 145. The positive correlation is not clear.  Delete "In addition to a few spots"

That sentence has been changed, and "in addition to a few spots" has been deleted.

Line 169. Delete the first "of".

The first "of" in line 169 has been deleted.

Line 284. Delete "of"

The "of" in line 284 has been deleted.

Figure 6. This figure is redundant. Delete it or delete Figure 3c.

Figure 6 in the original manuscript has been deleted.

Line 309. Replace "in addition to" with "Except for"

Response: already replaced

Line 311. Replace "precipitated" with "precipitation"

Response: already replaced

Reviewer 3 Report

A think that the article “S and Sr isotope compositions and trace element compositions of the Middle Jurassic evaporites in eastern Tibet: provenance and palaeogeographic implications” is quite interesting and could be published in the journal of “Minerals”. The authors study a specific region, with not many samples, but the methodology, interpretations, discussion are very interesting for the international scientific community. However, before publication, I think that some improvements could be introduced. Perhaps some aspects could be clarified or better explained.

- Introduction

 I suggest rewriting the first part of the introduction to better organize the ideas (lines 37-47). Authors start by talking about the Qiangtang basin as a reference, but the study basin is the Qamdo basin. Wouldn't it be better to talk about the Qamdo basin, its interest and then relate it to the Qiangtang basin? Similarly, when the Dongdaqiao formation is described as having marine and non-marine features with bivalves, then it is said the formation has vertebrates and fossil plants. Maybe, to indicate that there are ideas from different works, which show the diversity of deposits in different sedimentary environments.

 - Geological setting

 To better understand these deposits, it is better to put the stratigraphic sequence at the beginning, i.e., Fig. 9 as Fig. 2. These columns show better that the deposits of the Qamdo basin formation are continental with evaporites.

Explain with this figure the interest and relationship with the Qiangtang basin.

 - Methology

 It is interesting from a sedimentological point of view the presence of these lenticular gypsum in sandstones (line 94). Wouldn't it be possible to put a larger picture, to see better these evaporites in figure 2A?

 - Results

 In the text (line 135-136), the values of Rumei are from 0.707602 to 0.70854, but in the table I for the Rumei these values are from 0.707602 to 0.708163. Is it correct?

 - Discussion

 Probably the table 4 will be Fig 4 (line 184). In the caption, please, explain a little of the interest of the figure, e.g. what is the meaning of the numbers 1 and 2?

 I think it is also interesting to include in the discussion that in the Epicontinental Triassic of Europe there are several formations with detritic-coastal deposits that present layers of gypsum of marine origin (e.g., Ortí et al. 2017) (this is for the consideration of the authors).

 Reference: Ortí, F., Pérez-López, A., Salvany, J.M., 2017. Triassic evaporites of Iberia: Sedimentological and palaeogeograpical implications for the western Neotethys evolution during the Middle Triassic-Earliest Jurassic. Palaeogeography, Palaeoclimatology, Palaeoecology 471, 157–180.

 - Conclusions

 One question, in point 2 of the conclusions could some mention be made of the influence of Upper Triassic evaporites? (lines 401-403) Also, perhaps in order not to confuse, an explanation can be added in the line 403 explaining that continental water does not affect sulfates although it has a continental influence in its origin.

 Also, it is interesting to mention the celestina in point 2 (line 404-405)

 In the conclusions, please, do not abbreviate “Bacterial sulphate reduction” with the letters BSR (line 404), for readers who only read the conclusions, so that they can understand it.

Author Response

A think that the article “S and Sr isotope compositions and trace element compositions of the Middle Jurassic evaporites in eastern Tibet: provenance and palaeogeographic implications” is quite interesting and could be published in the journal of “Minerals”. The authors study a specific region, with not many samples, but the methodology, interpretations, discussion are very interesting for the international scientific community. However, before publication, I think that some improvements could be introduced. Perhaps some aspects could be clarified or better explained.

- Introduction

 I suggest rewriting the first part of the introduction to better organize the ideas (lines 37-47). Authors start by talking about the Qiangtang basin as a reference, but the study basin is the Qamdo basin. Wouldn't it be better to talk about the Qamdo basin, its interest and then relate it to the Qiangtang basin? Similarly, when the Dongdaqiao formation is described as having marine and non-marine features with bivalves, then it is said the formation has vertebrates and fossil plants. Maybe, to indicate that there are ideas from different works, which show the diversity of deposits in different sedimentary environments.

Response: It is pointed out that an integrated basin contained both Qiangtang and Qamdo basins accumulated sediments since Middle Jurassic, i.e., Qamdo basin is the eastern part of Qiangtang-Qamdo block. It is proposed that the discrepancy of different sedimentary environments reflected by bivalve assemblage and continental vertebrates and plants of the Dongdaqiao Formation was likely caused by a complex sedimentary process influenced by marine and non-marine inputs.

 - Geological setting

 To better understand these deposits, it is better to put the stratigraphic sequence at the beginning, i.e., Fig. 9 as Fig. 2. These columns show better that the deposits of the Qamdo basin formation are continental with evaporites. Explain with this figure the interest and relationship with the Qiangtang basin.

Response: This part has been transferred to the geological setting, and the comparison of those two lithologies have been made to illustrate the relationships and changes of Middle Jurassic sedimentary rocks from different basins.

 - Methology

 It is interesting from a sedimentological point of view the presence of these lenticular gypsum in sandstones (line 94). Wouldn't it be possible to put a larger picture, to see better these evaporites in figure 2A?

Response: It is a pity that we only took this only limited "panorama" (original Fig. 2A) of Rumei lenticular gypsum. Furthermore, the contact between sandstone and lenticular gypsum is ambiguous because of weathering. We attempt to take more photos with detailed information in the future.

 - Results

 In the text (line 135-136), the values of Rumei are from 0.707602 to 0.70854, but in the table I for the Rumei these values are from 0.707602 to 0.708163. Is it correct?

Response: Thank you for your correction, we misread 0.707854 as 0.70854. We already changed it in the text.

 - Discussion

 Probably the table 4 will be Fig 4 (line 184). In the caption, please, explain a little of the interest of the figure, e.g. what is the meaning of the numbers 1 and 2?

Response: Again, thank you for your correction, Table 4 already replaced by Figure 4. Number 1 and 2 in the Figure 4 caption have been explained by process 1 and process 2

 I think it is also interesting to include in the discussion that in the Epicontinental Triassic of Europe there are several formations with detritic-coastal deposits that present layers of gypsum of marine origin (e.g., Ortí et al. 2017) (this is for the consideration of the authors). Reference: Ortí, F., Pérez-López, A., Salvany, J.M., 2017. Triassic evaporites of Iberia: Sedimentological and palaeogeograpical implications for the western Neotethys evolution during the Middle Triassic-Earliest Jurassic. Palaeogeography, Palaeoclimatology, Palaeoecology 471, 157–180.

Response: Thank you for you notice, the paper you mentioned enlightened us about the paleogeographic configuration of the Middle Jurassic Qiangtang-Qamdo Basin. we already discussed this scenario based on this reference.

 - Conclusions

 One question, in point 2 of the conclusions could some mention be made of the influence of Upper Triassic evaporites? (lines 401-403) Also, perhaps in order not to confuse, an explanation can be added in the line 403 explaining that continental water does not affect sulfates although it has a continental influence in its origin.Also, it is interesting to mention the celestina in point 2 (line 404-405)

Response: The influence of Triassic evaporite recycling has been added and the occurrence of celestites is being mentioned in point 2.

 In the conclusions, please, do not abbreviate “Bacterial sulphate reduction” with the letters BSR (line 404), for readers who only read the conclusions, so that they can understand it.

Response: Already replaced the BSR by bacterial sulfate reduction.

Round 2

Reviewer 2 Report

The manuscript by Jinna Fei and others has been significantly improved and in my opinion is just about ready for publication.  I still find the abstract a bit confusing as pertaining to the conflicting relative roles of the S and Sr isotope compositions in determining the provenance of the evaporites.  However, the text is much less confusing and has become a useful and interesting paper.